# Enhancing PPB Affinity Prediction through Data Integration and Feature Alignment: Approaching Structural Model Performance with Sequences

## Abstract

One key step of protein drug development is the screening of protein-protein binding (PPB) affinity. The current mainstream screening method of PPB affinity is laboratory experiments, which are costly and time-consuming, making it difficult to quickly perform high-throughput screening. Various deep learning methods have been proposed to predict PPB affinity, but they are often limited by the availability of high-quality data and the compatibility of the algorithms with that data. In this work, we developed two AI models, PPBind-3D and PPBind-1D, to predict PPB affinity. PPBind-3D leverages structural information near the protein-protein binding interface to make its predictions. By employing monotonic neural network-constrained multi-task learning (MMTL), we effectively utilized heterogeneous affinity data from diverse wet lab experiments to expand the development dataset to over 23,000 samples, thereby enhancing the model's generalization capabilities. Additionally, PPBind-1D was developed using sequence data to address the lack of structural data in practical applications. During the training of PPBind-1D, we aligned it with PPBind-3D by incorporating an additional 42,108 no-affinity-label samples through an alignment approach. Finally, we demonstrated three application cases of our AI models in the virtual screening of protein drugs, illustrating that our models can significantly facilitate high-throughput screening.

## 1 Introduction

A critical challenge in the engineering of protein drugs is to assess the strength of binding between the protein drug and the target protein, known as protein-protein binding (PPB) affinity. The therapeutic effect of protein drugs typically relies on their ability to bind to specific target proteins. Protein drugs with high PPB affinity can bind more effectively to target proteins, thereby exerting a therapeutic effect. On the other hand, protein drugs with high PPB affinity can bind more specifically to the target proteins, reducing the impact on non-target proteins. This helps to reduce the side effects of the drug and enhance the safety of treatment.

High-throughput PPB affinity screening can accelerate the development of protein drugs. In recent years, technologies such as protein microarrays(MacBeath, 2002) and the Octet system(Cameron et al., 2021) have been developed. Although these experimental methods are accurate, they require cumbersome experimental operations, strict experimental conditions, and expensive equipment and consumables. Therefore, algorithm-based PPB affinity prediction is a more promising paradigm for high-throughput screening.

However, the PPB affinity prediction is limited by the generalization of the algorithm model, which often lacks more diverse and high-quality data(Kortemme, 2024). On the other hand, in real-world scenario, the accurate true-structure of the mutant-type complex is usually unavailable. These limitations highlight the need for continued development and refinement of computational methods to improve the efficiency and accuracy of PPB affinity screening.

In order to address the aforementioned challenges, this work makes three contributions. First, the largest protein affinity dataset to date, PPB-Affinity(Liu, 2024), comprising 12,062 samples, was employed in the development of a geometric deep learning model, PPBind-3D, which predicts PPB affinity based on structural features near the binding interface of protein-protein complexes. We also trained the model by integrating the heterogeneous affinity data, especially Deep Mutation Screening(DMS)(Fowler & Fields, 2014) data, through a monotonic neural network module(Sill, 1997; Wang et al., 2023), thereby further enhancing the model's generalization performance. Second, we proposed a more rigorous method for clustering protein complex structures. In previous studies of AI-predicted binding affinity, there has always been data leakage of varying degrees due to the lack of rigorous data division, making it impossible to accurately assess the predictive performance of the model. To address this, we calculated the features of protein complex structures in a SE(3)-Invariant manner using the iDist algorithm(Bushuiev et al., 2023) and then clustered the protein complex structure features based on graph partition algorithms(Karypis & Kumar, 1998), thus achieving a more rigorous data division. Finally, we developed a sequence model PPBind-1D based on our innovative "Feature Alignment" principle, which guided the sequence model through structural models to achieve the predictive performance of structural models. Additionally, a substantial number of authentic protein complex structures were employed, including unlabeled samples, to assist in training the PPBind-1D model to align with the PPBind-3D model.

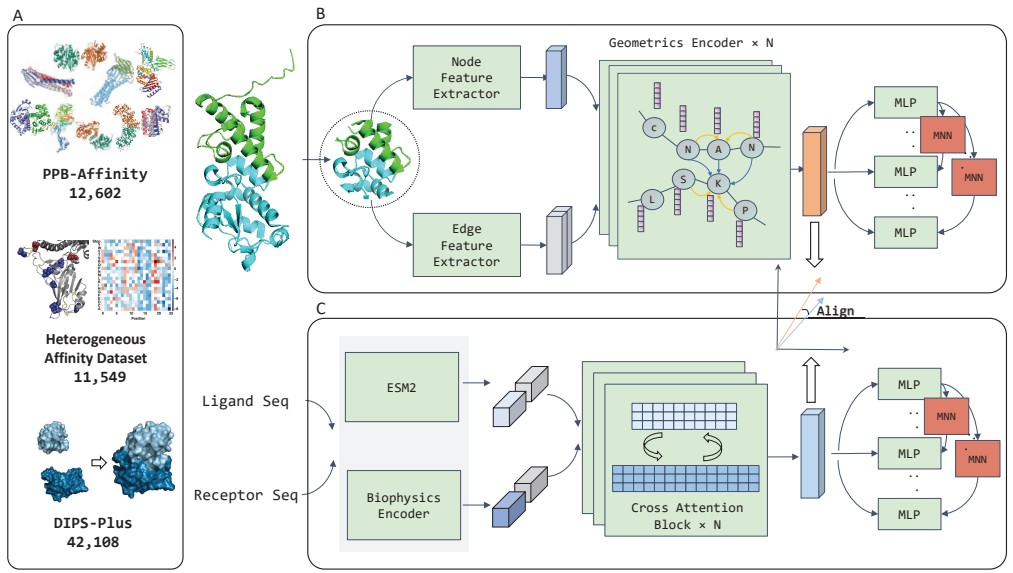

Figure 1: (A) Dataset. (B)PPBind-3D. (C)PPBind-1D.

## 2 RELATED WORK

**Molecular dynamics based methods.** Representative methods include, Rosetta Flex ddG(Kellogg et al., 2011), FoldX(Schymkowitz et al., 2005), GROMACS(Abraham et al., 2015), are based on physical principles. They predict free energy and its changes by analyzing and evaluating factors such as chemical bonds, residue conformation, Coulomb forces, van der Waals forces, and thermodynamic integration, offering good generality. However, these methods require complex computational processes, leading to high demands for computational resources and longer calculation times. More importantly, they often have limitations in the prediction accuracy of PPB affinity and typically require known three-dimensional structures of complexes, making it difficult to apply them to high-throughput virtual screening of PPB affinity.

**AI algorithms for predicting binding free energy change upon mutation($\Delta\Delta$G).** Representative algorithms include TopNetTree(Wang et al., 2020), ddGpred(Shan et al., 2022), RDE-Net(Luo et al., 2023) and UniBind(Wang et al., 2023), etc. These algorithms are mainly applied to affinity matu-

ration, where a few mutations are made at specific sites of the parent protein to enhance its binding affinity with the receptor protein. However, the limitation of these methods is the inability to predict the affinity changes resulting from animo acid deletion and insertion, which restricts its application in virtual screening of proteins with varying lengths.

**AI algorithms for predicting binding free energy change($\Delta$G).** Representative methods include CSM-AB(Myung et al., 2020a; 2022; 2020b), PPI-Affinity(Romero-Molina et al., 2022b), AREA-AFFINITY(Yang et al., 2023a;b), and DG-Affinity(Yuan et al., 2023). These methods extract features of the three-dimensional structure of protein complexes and the amino acid sequence of proteins in order to predict the affinity. Specifically, in PPI-Affinity, the spatial structure of residues is grouped, and topographic indices, thermodynamic indices, property-based indices, and other features are reconstructed and calculated through aggregation operators to obtain features with spatial information(Ruiz-Blanco et al., 2015). In AREA-AFFINITY, the area of interface residue pairs is first calculated, and dr-sasa is used to obtain surface area. Then information such as amino acid types and physicochemical properties at the interface and surface are aggregated to obtain features with three-dimensional structural information. Despite the inclusion of spatial structural information in the extracted features, the three-dimensional structure of the protein complex is not explicitly described, and well-defined features are more conducive to the learning of AI models.

## 3 DATASET

### 3.1 DATA COMPOSITION

Our data source is shown in Fig.1(A), which mainly consists of four parts: (1) **PPB-Affinity Dataset:** This is the largest protein affinity dataset to date, where each sample has a experimentally measured $\Delta$G value, the three-dimensional structure of the wild-type complex, and mutation information, etc. (2) **Heterogeneous DMS Affinity Datasets:** Heterogeneous affinity datasets such as PBAD-AS(Chan et al., 2020), PDAD-SA(Starr et al., 2022), where the affinity measurements are not $\Delta$G or dissociation constant($K_D$) values, but rather $K_{d,app}$ or $log_2$ enrichment ratio. Within the same set of experiments, these measurements are positively or negatively correlated with the affinity $\Delta$G values, but they cannot be directly converted to $\Delta$G values using known formulas. (3)**Protein Complex Structure Dataset:** DIPS-Plus(Morehead et al., 2023), an enhanced, feature-rich dataset of 42,108 complexes for geometric deep learning of protein interfaces. (4)**Validation Case Dataset:** Affinity data of nanobodies with different antigens, including CTLA-4, PD1, PD-L1, and HEL.

The PPB-Affinity dataset and the heterogeneous DMS affinity datasets are used for the development and validation of PPBind-3D and PPBind-1D, while the DIPS-Plus dataset is used exclusively for the development of PPBind-1D. The validation case Dataset does not participate in model training.

### 3.2 DATA PARTITIONING

Data partition is usually used to verify the true performance of the model. For protein affinity data, however, traditional random partition is not reasonable because the same or similar protein complexes may appear in both the training set and the validation set, resulting in an inability to correctly evaluate the model's performance. Luo Shitong(Luo et al., 2023) proposed data partition based on PDB code, but there may be data leakage(Bushuiev et al., 2023) due to the fact that protein complexes with different PDB codes may also be composed of homologous proteins (such as 2NU0, 1SGQ). In order to address this issue, we propose a novel data partitioning method based on Anton Bushuiev's SE (3) PPI redundancy removal technique iDist(Bushuiev et al., 2023). This method has the advantage of less data leakage and is more conducive to reflecting the true of the model.

First, we computed the similarity of all PDB files in the PPB-Affinity dataset using iDist, and employed the nearest neighbors algorithm to identify several most similar complexes for each complex. Treating each complex as a node and connecting similar complexes with edges, we could represent the similarity relationships of the dataset as a Graph. Next, the graph partitioning algorithm METIS(Karypis & Kumar, 1998) was applied to divide the dataset into N folds for cross validation of the proposed models. Finally, we set N to 5 in our experiments and used the Fruchterman-Reingold algorithm to arrange the nodes to visualise the graph as shown in Fig.2. Nodes lacking edge connections constitute the "ring" in the figure. Conversely, nodes with a greater number of

edge connections will be situated in closer proximity to the "centre" of the circle. We use different colours to represent different data folds, and it is evident that each data fold exhibits distinct characteristics. Optimising the partitioning quality through minimising edge cutting by METIS, it is possible to group together nodes with greater similarity. This approach to data partitioning facilitates the enrichment of homologous or similar structures within a single data fold. Furthermore, we also analyzed the differences in data partitioning methods in A.3.

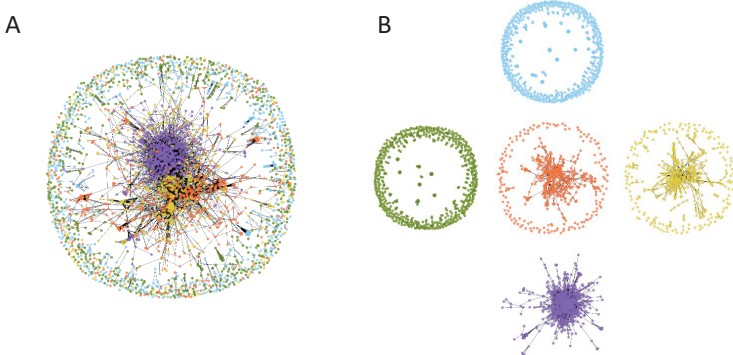

Figure 2: (A): Overall rendering divided into five parts. (B): Five subplots that make up the overall rendering

## 4  PPBIND

### 4.1  PPBIND-3D

We designed the network as illustrated in Fig.1(B). Firstly, it should be noted that despite the significant differences in protein length and conformation observed between different protein complexes, they all possess a binding interface that directly affects affinity. In order to concentrate the model on the area in close proximity to the binding interface, for each amino acid residue present in the receptor of the protein complex, if there is an amino acid residue present on the ligand and the distance between their C-alpha atoms is less than 10 Å, then this pair of residues is defined as the binding sites. The amino acid residues in the receptor and ligand in closest proximity to the binding site were extracted using the K nearest neighbour algorithm, which identified the visible patches of the model. We employ ROTAMER DENSITY ESTIMATOR(Luo et al., 2023) to extract and simulate the amino acid side chain potential conformational distribution information. To fully leverage the affinity data of various protein complex mutations without the necessity of inputting mutant structures, thereby significantly expanding the quantity of available data.

Subsequently, in order to fully leverage the information derived from the three-dimensional structure, we represented the residues as nodes and their pairs as edges. This allows us to represent the protein complex as a complete graph. Specifically, we define the node feature vector at the residue level as $h$, which includes the type of amino acid residue, physicochemical properties of amino acids, relative solvent-accessible surface area, types of dihedral angles, and types of side chain torsion angles. The edge feature vector is denoted as $e$, including the amino acids types, differences in relative solvent-accessible surface areas, relative positions, Euclidean distances, and virtual dihedral angles between the two connected residues.

The core of our architecture is the geometric encoder, inspired by DDG-Pred(Shan et al., 2022) and RDE-Network(Luo et al., 2023), which is an SE (3) invariant attention module. In the Geometric Encoder, two modes of feature updating, 'SELF' and 'MUTUAL' , are designed. The 'SELF' mode involves the ligand or receptor updating features based solely on its own structural information, while the 'MUTUAL' mode involves the ligand or receptor updating features based on the structural information of the counterpart. A complete feature update is defined as a process that begins with 'SELF' and then proceeds to 'MUTUAL' once more. Specifically, for a $L$-layer model, the attention computation process in the $l$-th ($1 \leq l \leq L$) layer Geometric encoder can be represented as follows:

$$\alpha_{ij}^{h(l)} = \frac{1}{\sqrt{d}} \operatorname{Linear}\left(h_i^{(l)}\right) \cdot \operatorname{Linear}\left(h_j^{(l)}\right)^T \tag{1}$$

$$\alpha_{ij}^{e(l)} = \operatorname{Linear}\left(e_{ij}^{(l)}\right) \tag{2}$$

$$\alpha_{ij}^{spatial(l)} = \gamma \left\| \left(R_i \operatorname{Linear}\left(h_i^{(l)}\right) + T_i\right) - \left(R_j \operatorname{Linear}\left(h_j^{(l)}\right) + T_j\right) \right\|_2 \tag{3}$$

$$\alpha_{ij}^{(l)} = \operatorname{softmax}\left(\alpha_{ij}^{h(l)} + \alpha_{ij}^{e(l)} + \alpha_{ij}^{spatial(l)}\right) \tag{4}$$

Among them, $R$ and $T$ represent the rotation matrix and translation vector of the $i$-th residue transformed from the local coordinate system to the global coordinate system; $h$ represents the node feature; $\gamma$ is a learnable parameter, and $\alpha_{ij}^{(l)}$ is the weight of the $l$-th layer Geometric encoder attention. In the "SELF" mode, both $i$ and $j$ are residues in the ligand or receptor. If $i$ and $j$ are not homologous, the mode is "MUTUAL". The process of feature updating can be expressed as:

$$h_i^{(l)\prime} = \operatorname{Concat}\left(\alpha_{ij}^{(l)} \operatorname{Linear}\left(h_i^{(l)}\right), \sum_j \alpha_{ij}^{(l)} \operatorname{Linear}\left(e_{ij}^{(l)}\right), \ R_i^{-1}\alpha_{ij}^{(l)} \operatorname{Linear}\left(h_i^{(l)}\right) - T_i\right) \tag{5}$$

Many studies on protein-protein binding affinities, such as the heterogeneous DMS affinity dataset we collected, did not measure the $K_D$ or $\Delta$G values directly, but measured values like ligand enrichment, which were more abundant. Although these values cannot be directly converted into $K_D$ or $\Delta$G values through a formula, they are positively or negatively correlated with $K_D$ and $\Delta$G. To leverage this valuable heterogeneous affinity wet-lab data, we referred to G. Wang's work(Wehenkel & Louppe, 2019) and introduced Monotonic Neural Networks into the prediction head, which we called it as monotonic neural network-constrained multi-task learning (MMTL). Specifically, affinity prediction is treated as multi-task learning, with each task corresponding to a distinct prediction head, all prediction head sharing a common backbone network. The primary prediction head is tasked with predicting $\Delta$G values, while the other prediction heads predict various non-$\Delta$G from different sources. Thus, the learning objective for PPBind-3D can be expressed as a minimization objective function as follows.

$$\arg\min_{\theta,\theta_t}\left\{\frac{1}{T^*N}\sum_{t=1}^{T}\sum_{i=1}^{N}(y_{t,i} - M_\theta(x_i)_t)^2 + \frac{1}{(T-1)^*N}\sum_{t\neq 1}^{T}\sum_{i=1}^{N}(\lambda_t \cdot M_\theta(x_i)_1 - F_{\theta_t}(\lambda_t \cdot M_\theta(x_i)_t))^2\right\} \tag{6}$$

Here, $T$ represents the task (prediction head) index, $y_{t,i}$ represents the true value of the $i$-th sample for task $t$. $M_\theta$ is PPBind network used to predict values for different tasks. $F_\theta$ is a neural network that approximates the computation of integrals using the Crenshaw-Coulters method, thereby enhancing the accuracy of the integrals. It is capable of learning and integrating monotonically increasing functions. For more details on neural network $F_\theta$, please refer to A.4. As $F_\theta$ is applicable solely to functions that control monotonically increasing functions, we employ the symbol $\lambda_t$ to denote the monotonicity of task label values with respect to $\Delta$G, where a value of 1 denotes monotonically increasing and -1 denotes monotonically decreasing. The term before the + is the mean squared error formula, which is used for training the model to predict affinity values. The term following the + is used to train the model on the monotonicity between different affinity metrics and the $\Delta$G values.

## 4.2 PPBIND-1D

In protein complexes, there are often more than one chain of ligands and receptors, i.e., the ligand and receptor themselves might constitute a complex. Currently, the protein language models or other sequence models can only take monomeric sequences and a linker is commonly used to connect the complex sequences into a single entity to accommodate complex sequences to handle complex sequences. To simplify the problem, this study considers data where the number of receptor and ligand chains does not exceed two. Thus, the most complex protein complex situation addressed

here is that both the ligand and receptor are dimers. A linker consisting of 25 Gly residues is used to connect the sub-complexes of the ligand or receptor.

We designed the network as illustrated in Fig.1(C). We used physicochemical properties of amino acids and protein language models ESM2(Lin et al., 2022) to extract the basic features of ligand sequences or receptor sequences, respectively. Next, we simulated the process of protein-protein interactions using a cross-attention mechanism to facilitate information transferring and updating between ligand and receptor. The cross attention mechanism can be represented by the following formula:

$$\beta_{li,rj} = \text{softmax}\left(\frac{1}{\sqrt{d}}\text{Linear}\left(s_{li}\right) \cdot \text{Linear}\left(s_{rj}\right)^T\right) \tag{7}$$

$$h'_{li} = \beta_{li,rj}\text{Linear}\left(s_{li}\right) \tag{8}$$

Where $s$ represents sequence features, $\beta_{li,rj}$ represents the attention of the $i$-th residue in the ligand to the $j$-th residue in the receptor, and similarly, the attention of the $i$-th residue in the receptor to the $j$-th residue in the ligand can be expressed as $\beta_{ri,lj}$.

In order to enable the sequence model to learn structural information, we proposed a novel Alignment method for training the model that was more lightweight and also simplified the model training process, allowing the model to extract as much structure-related features as possible and to approach the data distribution of the latent vector in the structural model more closely. "Alignment" consist of cosine similarity and mean square error, defined as:

$$L_{\text{align}} = \frac{\chi_{\text{structure}} \cdot \chi_{\text{sequence}}}{\max\left(\|\chi_{\text{structure}}\|_2, \epsilon\right) \cdot \max\left(\|\chi_{\text{sequence}}\|_{2'}\epsilon\right)} + \left(\chi_{\text{structure}} - \chi_{\text{sequence}}\right)^2 \tag{9}$$

Where $\chi$ is the feature vector before inputting into the multi-modal prediction head. The purpose of this design is to ensure that the direction of the feature vectors is as uniform as possible and the magnitude of the modulus is close, with features extracted solely from sequence information aligning with those extracted from structural information, thereby enhancing the predictive performance of the sequence model. To provide further guidance to Alignment, the architecture and weights of the multimodal prediction head of PPBind-3D are transferred to PPBind-1D. The learning objective of PPBind-1D can be defined as a minimization objective function as the sum of $L_{\text{align}}$ term and Equation(6).

## 5 RESULT

### 5.1 EVALUATION

We trained and tested PPBind-3D by the PPB-Affinity dataset and the DMS-Het dataset, where the DMS-Het dataset was for model training only and the PPB-Affinity dataset was for cross-validation. Under strict data partitioning, the five-fold cross-validation performance of PPBind-3D on the PPB-Affinity dataset was showed as Fig.3A. Fig.3B illustrates the performance of PPBind-3D when trained and tested at an 8:2 ratio with random partitioning.

Similarly, PPBind-1D has been validated using both strict and random partitioning, in a manner consistent with the validation of PPBind-3D. However, the training set for PPBind-1D additionally included DIPS-Plus. After a simple filtering of the PPB-Affinity dataset (as described in section '4.2 PPBind-1D'), PPBind-1D was trained based on the principle of sequence-structure-alignment. The performance of PPBind-1D was as follows in Fig.4.

The test metrics for random partitioning are significantly higher than those for strict partitioning. This is because the random partitioning introduces data leakage, which artificially boosts the test metrics. In contrast, strict partitioning avoids data leakage and provides a more accurate evaluation of the model's generalization performance. These results demonstrate the superiority of our proposed data partitioning method. Furthermore, the Pearson and Spearman correlations of our PPBind-3D and PPBind-1D models are both greater than 0.6 under the strict partitioning of data, indicating that our model architecture is preeminent.

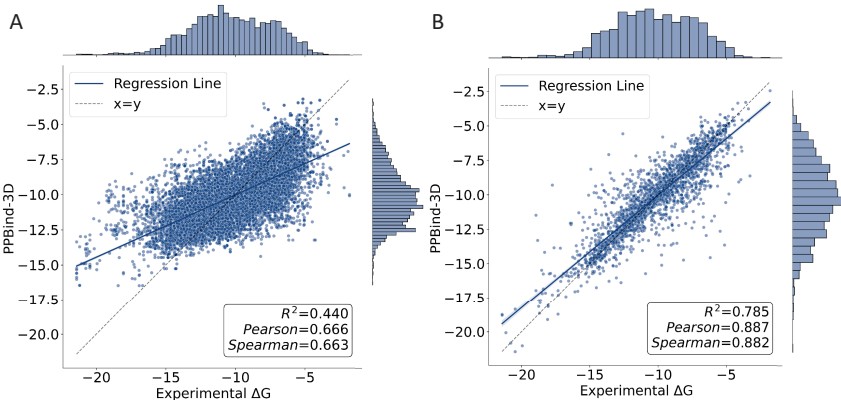

Figure 3: (A)The performance of PPBind-3D under strict data partitioning.(B)The performance of PPBind-3D under random data partitioning.

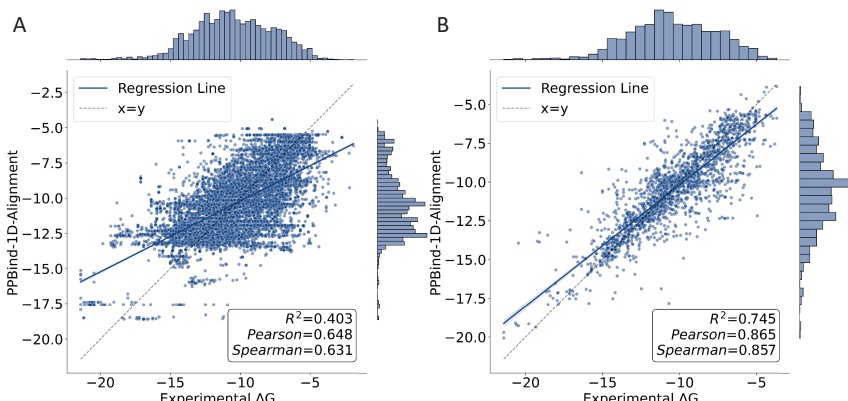

Figure 4: (A)The performance of PPBind-1D under strict data partitioning.(B)The performance of PPBind-1D under random data partitioning.

To better evaluate the performance of our model, we used the following models as baseline comparisons: PRODIGY(Xue et al., 2016), which predicts affinity based on intermolecular contacts and properties derived from non-interface surfaces; DFIRE(Liu et al., 2004), which predicts affinity based on a potential function using the ideal gas state as a physical reference; CP_PIE(Ravikant & Elber, 2010), a mathematical programming-based approach for protein-protein docking filtering and scoring that utilizes residue contacts and overlap areas; ISLAND(Abbasi et al., 2020), which employs sequence-based features and a machine learning model to predict affinity; and ProBAN(Bogdanova & Novoseletsky, 2024), which utilizes complex structural data and a deep 3D convolutional neural network to predict affinity. The test data and baseline model metrics were sourced from ProBAN. The test data consists of two components: test set 1, which includes 126 samples, and test set 2, which includes 83 samples, with all samples in set 2 being protein complexes composed of two chains. Both sets are subsets of those in PDBbind v2020(Wang et al., 2004). Additionally, all PDB entries identified in the test data were excluded, and PPBind-3D was retrained. The resulting performance are illustrated in Table 1, from which it can be seen that PPBind-3D outperforms other algorithms in all aspects, demonstrating its superior performance.

## 5.2 VISUALIZATION OF ALIGNMENT

To observe the effectiveness of 'Alignment', the feature representations are visualised by dimensionality reduction using the following steps:

Table 1: Comparison between PPBind-3D and other models

| Method | Test set 1(126) | | | Test set 2(83) | | |
|---|---|---|---|---|---|---|
| | Pearson | MAE(kcal/mol) | RMSE(kcal/mol) | Pearson | MAE(kcal/mol) | RMSE(kcal/mol) |
| PRODIGY | - | - | - | 0.28 | 2.47 | 3.52 |
| DFIRE | - | - | - | 0.08 | 25.05 | 29.17 |
| CP_PIE | - | - | - | -0.10 | 10.90 | 11.27 |
| ISLAND | - | - | - | 0.28 | 2.30 | 2.85 |
| PPI-Affinity | - | - | - | 0.49 | 1.83 | 2.40 |
| ProBAN | 0.60 | 1.60 | 1.95 | 0.55 | 1.75 | 2.28 |
| PPBind-3D(ours) | **0.626** | **1.482** | **1.898** | **0.559** | **1.647** | **2.210** |

**Step1** Extract high-dimensional feature representations of the training samples using the PPBind-3D model, and fit a dimensionality reduction function $F_U$ using the UMAP (Uniform Manifold Approximation and Projection) algorithm.

**Step2** Extract high-dimensional feature representations of the validation samples using the PDBind-3D, PDBind-1D, and PPBind-1D-w/o Align models, respectively.

**Step3** Individually project the three sets of high-dimensional feature representations onto a 2D plane using the fitted $F_U$ function and visualize them.

By comparing the three dimensionality reduction visualizations Fig5, it can be observed that the dimensionality reduction representation of the PPBind-1D model retains a similar data topological structure to that of the PPBind-3D model, whereas the PPBind-1D model without 'Alignment' exhibits a scattered state. This indicates that the representations extracted by the PPBind-1D model are similar to those of the PPBind-3D model, suggesting that it is possible to enhance the prediction accuracy of the PPBind-1D model to the level of the PPBind-3D model through our proposed 'Alignment' method.

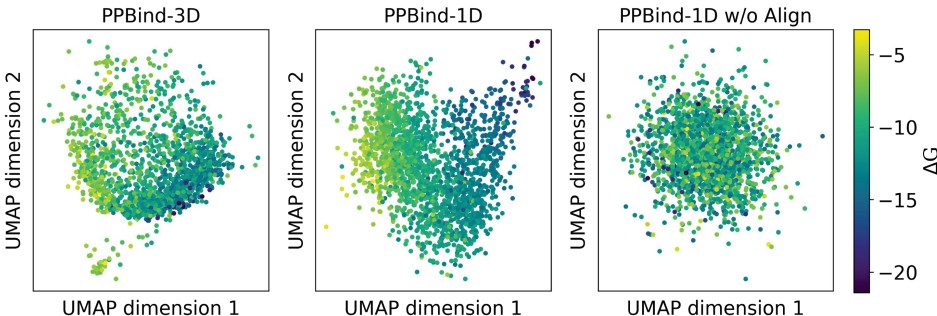

Figure 5: Visualize the representation of three models

## 5.3 VIRTUAL SCREENING

To validate the performance of the model in virtual screening of protein affinity, a series of three case studies was conducted. In order to enhance the precision of the screening outcomes, this section employs the models that has been trained through the random partitioning of the data set. At the same time, we also compared three cases with the training data, including the Euclidean distance represented by iDist, PDB ID and its descriptive information. For details, please refer to A.6.

**Case1. Based on PPBind-3D, predict affinity from real structure.** We have compiled a set of recent experimental data (Kang-Pettinger et al., 2023) on affinity and complex structures, which have not yet been included in the PPB-Affinity dataset. This dataset involves affinity $K_D$ values and complex structures for various antibodies binding to antigens such as Cytotoxic T-Lymphocyte Antigen 4 (CTLA-4), Programmed Death Protein 1 (PD-1), and Programmed Death-Ligand 1 (PD-L1), as well as their mutants. PPBind-3D, was used to predict the affinities of these antibody-antigen complexes. The Fig.6A is based on whether the affinity originates from a mutant or not, while the

Fig.6B is based on the PDB code. Overall, this case shows that PPBind-3D can be used for affinity prediction and virtual screening when real structural information is available.

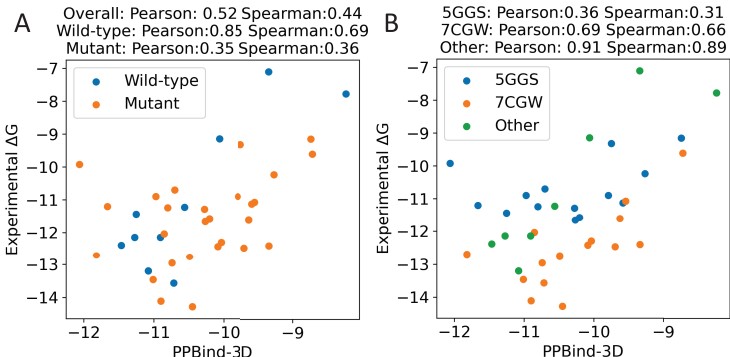

Figure 6: Display the prediction results of the model under real structure. (A) is based on the type, (B) is based on the PDB code.

**Case2. Anti-Hen Egg lysozyme antibodies affinity ranking.** To assess the performance of the proposed models in virtual affinity screening without real structures, we conducted case2. We obtained a set of 38 $K_D$ values for different nanobodies binding to HEL, as measured by Porebski et al. (2024) through experiments. We then predicted the complex structures of each nanobody with HEL using AlphaFold3, follwed by affinity predictions with PPBind-3D and PPBind-1D (Fig.7).

As shown in Fig7 A, the affinity predictions for structures using PPBind-3D based on AlphaFold3 were found to be of a comparable level to those using PPBind-1D-Align. Conversely, PPBind-1D-No-Align performed significantly less well than PPBind-1D-Align. It was observed that the predicted structures in this batch exhibited a general low ipTM(Fig.7D), indicating potential inaccuracy in the structure prediction of the interface region. Furthermore, it was determined that distinct complexes exhibit disparate epitopes(Fig.7E), which markedly influence the affinity strength and ultimately result in the failure of PPBind-3D prediction.

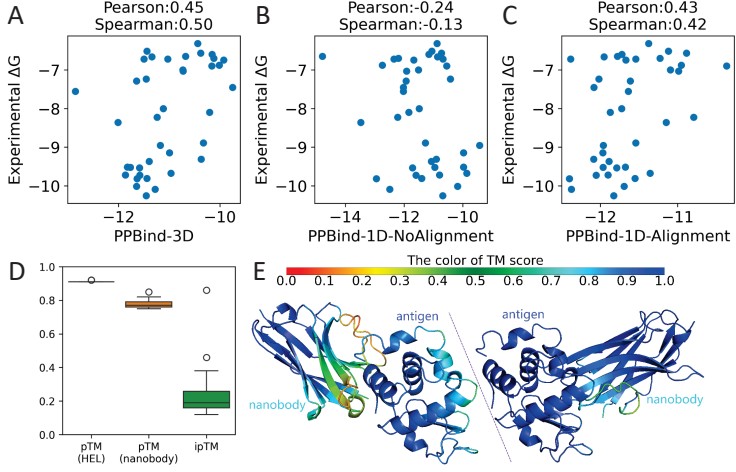

Figure 7: Correlation between the affinities predicted by (A) model PPBind-3D, (B) model PPBind-1D-NoAlignment, (C) model PPBind-1D-Alignment and the actual affiinties in Case 2; D. Box plot of the ipTM and pTM for the complex structures predicted by AlphaFold3 in Case 2; E. Two structures predicted by AlphaFold3 in case 2. The left side has lowest ipTM, and the right side has highest ipTM.

**Case3. Anti-PD-L1 antibodies affinity ranking.** The antibodies and the affinity values derived from Brzostek et al. (2016); Gao et al. (2020); Guan et al. (2023); He et al. (2017); Hong et al. (2021); Rajasekaran et al. (2024); Tan et al. (2018; 2017) were used to validate ours model in the case. Similar to Case 2, only the sequences of the proteins are known. We also used AlphaFold3 to predict complex structures and compared the predictions of the three models.

As shown in Fig.8A, the results of the predictive modelling demonstrate that PPBind-3D is the least effective; PPBind-1D-No-Align is the second-best performer, and PPBind-1D-Align is the most accurate. We found that even with high ipTM (Fig.8D), docking posture and epitopes and paratopes varied between individuals (Fig.8E), which we believe contributes to affinity prediction.

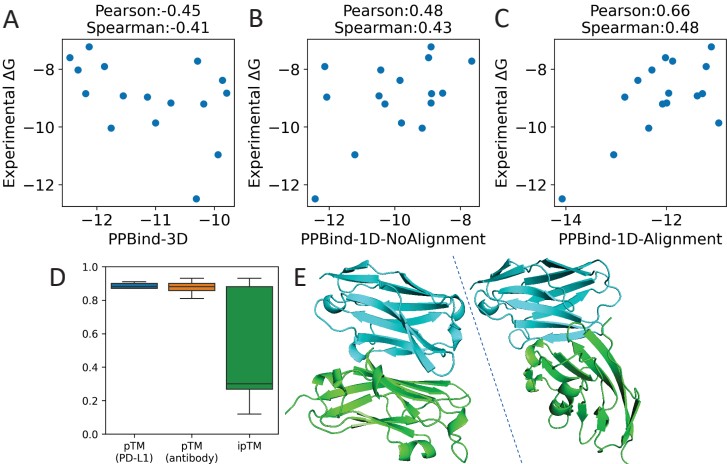

Figure 8: Correlation between the affinities predicted by (A) model PPBind-3D, (B) model PPBind-1D-NoAlignment, (C) model PPBind-1D-Alignment and the actual affinities in Case 3; D. Box plot of the ipTM and pTM for the complex structures predicted by AlphaFold3 in Case 3; E. The two structures for which AlphaFold3 predicted the highest ipTM scores in Case 3. The blue chains is PD-L1, and the green chains is antibody.

## 6   CONCLUSION

In this paper, a substantial corpus of disparate PPB affinity data was integrated, and a data partitioning method was proposed that can markedly diminish data leakage. The feasibility of this data partitioning method was demonstrated by training the model PPBind-3D. Subsequently, our model PPBind-1D was trained based on a novel training paradigm based on a principle of sequence-structure-alignment, which effectively combines the precision of structural models with the expediency of sequence models. The simulation of a genuine virtual screening scenario has demonstrated that PPBind-1D-Align is highly compatible with the actual application requirements.

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

# A APPENDIX

## A.1 REPRODUCIBILITY

The codes for our work are available at https://anonymous.4open.science/r/PPBind-for-ICLR2025

## A.2 DETAILS ON THE HYPERPARAMETERS

For training the PPBind models we use the Adam optimizer with an initial learning rate at 1e-4. We use a batch size of 16. We train the models for 10,000 to 300,000 iterations across various experiments. We also use the plateau learning rate scheduler for all model training. For PPBind-3D training, we used K nearest neighbour algorithm with K=64, to respectifully clip the amino acid residues in the receptor and ligand in closest proximity to the binding site. With batch size=16, using a single NVIDIA A100 GPU, training PPBind-3D for 100000 iterations takes about 7hours, and training PPBind-1D for 360,000 iterations takes about 22hours.

## A.3 COMPARE DIFFERENT WAYS OF PARTITIONING DATA

We evaluated the partition performance of three methods , namely the proposed partition method, partitioning according to PDB codes, and partitioning according to sample randomization. The minimum, maximum and average Euclidean distances between each fold of data were calculated shown as Fig.9. Observing the distribution of the minimum, there were similar complexes between different folds in both randomized divisions. In our proposed strict division method, there were no similar complexes between each fold. From the average distance plot, the two randomized division methods were compared with our proposed method, which divides as many similar complexes as possible in the same fold, because the average distance per fold of the randomized division method is very close to the average distance per fold of the randomized division method, whereas the average distance per fold of our method is somewhat different, and the value is both large and small.

## A.4 DETAILS OF MONOTONIC CONTROL

Monotonic Neural Networks (MMN) fundamentally represent a monotonic function $y = F(x, \theta)$, facilitating the transformation between two scalar values $x \in \mathbb{R}$ and $y \in \mathbb{R}$. This transformation, without loss of generality, strictly enforces a monotonic positive correlation between $x$ and $y$. Wehenkel & Louppe (2019) constructed such a monotonic function by integrating a strictly positive derivative $f(t, \theta)$, as expressed in the following equation:

$$F(x, \theta) = \int_0^x f(t, \theta) \, dt + F(0, \theta)$$

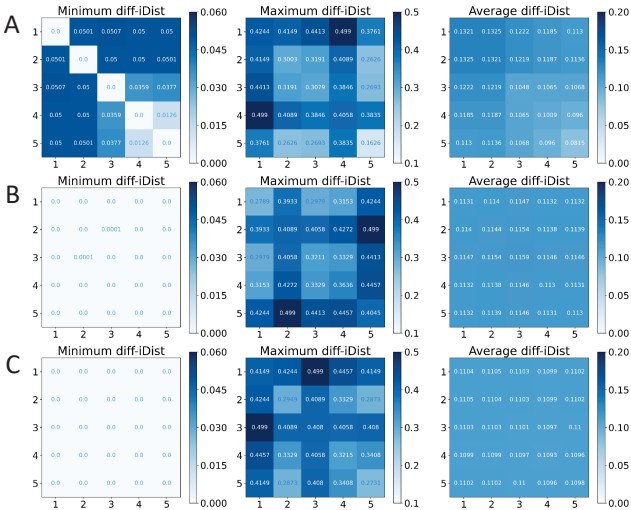

Figure 9: (A): Our proposed strict partitioning method; (B): Randomly partitioning according to PDB; (C): Randomly partitioning according to samples

Here, $f(t, \theta)$ is always greater than zero, and $F(0, \theta)$ is a constant. We represent $f(t, \theta)$ using a simple Multi-Layer Perceptron (MLP) network, ensuring that the output remains positive by applying a ELU activation function in the final layer and adding one to the network output value. Subsequently, we employ the Clenshaw-Curtis quadrature method for numerical integration to compute $y = F(x, \theta)$ over the interval $[0, x]$. In practical implementation, we can compute the forward integral and backward differentiation of $F$ more efficiently through mathematical transformations, with specific details available in the referenced Github link: https://github.com/AWehenkel/UMNN/blob/master/models/UMNN/MonotonicNN.py.

In our application case, the input $x$ to the monotonic neural network is the model-predicted $dG$ value multiplied by its sign, while the output $y$ corresponds to other heterogeneous affinity values. The sign indicates the monotonicity between the $dG$ value and the heterogeneous affinity values, where +1 denotes a monotonic positive correlation and -1 denotes a monotonic negative correlation. Specifically, the sign of $K_{d,app}$ is +1, while the sign of the log2 enrichment ratio is -1.

## A.5 ABLATION STUDY

To investigate the impact of data partitioning methods, network design, and training strategies on model performance, we conducted ablation experiments on the PPB-Affinity dataset, as summarized in the table2. The overall metrics were derived from the complete PPB-Affinity dataset, while the Per-Structure metrics were obtained from samples with more than 10 mutants in the PPB-Affinity dataset.

**The structural model PPBind-3D outperforms the sequence model PPBind-1D.** To rigorously assess the effects of strict versus random data partitioning on model performance, we performed 10 experiments. The strict partitioning employed five-fold cross-validation (details can be found in the Methods section under Data Partitioning), while the random partitioning used an 80:20 split between the training and validation sets without cross-validation. In the case of random data partitioning, PPBind-3D significantly outperformed PPBind-1D, particularly in the Per-Structure metrics. This aligns with the intuition that structural information is more beneficial for predicting binding affinity than sequence information, especially in capturing the affinity differences induced by mutations. Conversely, under strict data partitioning, both models exhibited a notable decline in performance. However, PPBind-3D still maintained superior performance over PPBind-1D, particularly in the Per-Structure metrics. This comparative analysis suggests that random data partitioning likely introduces data leakage, resulting in inflated performance evaluations to some extent. Conversely, it demonstrates that our proposed strict data partitioning method can substantially mitigate the risk of data leakage.

**MMTL enhances the models' generalization performance.** A comparison between Experiments 1 and 2 reveals that employing MMTL (utilizing the DMS-Het dataset) results in a significant improvement in overall model performance. This indirectly supports the reliability of the non-dG affinity data. Considering that incorporating more data for training is beneficial for improving the model's generalization performance, we advocate for the use of MMTL.

**S&M-attention outperforms All-attention.** In the Geometric Module, we conducted ablation studies (Experiments 2, 3) to analyze the impact of different attention mechanisms. "All-attention" refers to a method that does not distinguish between the receptor and ligand, treating the complex structure as a whole for attention calculation and feature updating. In contrast, "S&M-attention" (self-attention and mutual attention) treats the receptor and ligand as individual entities. It first computes self-attention within each entity to update their features, followed by mutual attention to capture interactions between the receptor and ligand, further refining their respective feature representations. By comparing Experiments 2 and 3, it is clear that S&M-attention significantly outperforms All-attention.

**The alignment mechanism enhances PPBind-1D, and incorporating unlabeled samples (DIPS-Plus dataset) further boosts model performance.** To validate the effectiveness of our proposed "Alignment" method for PPBind-1D, we conducted Experiments 4,5,6 and 8,9,10. In Experiments 6 and 10, the models were trained directly without using Alignment. Experiments 5 and 9 employed the Alignment method but did not utilize unlabeled samples. Experiments 4 and 5 aligned with Experiment 3, while Experiments 8 and 9 aligned with Experiment 7. Under strict data partitioning (Experiments 4, 5, and 6), it is evident that models using the Alignment mechanism outperform those trained directly across all metrics. Additionally, incorporating unlabeled samples further improves the model's performance, bringing it closer to PPBind-3D. In contrast, experiments 8, 9, 10 show that the model incorporating unlabeled samples for alignment performed the worst. This is due to data leakage between the PPB-Affinity training and test sets under random data partitioning, leading to inflated test set performance.

## A.6 COMPARISON OF CASE DATA

In order to investigate the potential correlation between the three validation cases and the training data, we employed the iDist method to characterise all the samples. We then computed and identified the training data PDB with the smallest Euclidean distance from the case data and obtained brief descriptions of these by querying the RCSB. The above information was then collated into Tables 3, 4, 5 and 6

Table 2: The result of Ablation Study

| Index | Data split | network | Overall | | | | Per-Structure | |
|---|---|---|---|---|---|---|---|---|
| | | | Pearson | Spearman | R2 | MAE | Pearson | Spearman |
| 1 | strict | PPBind-3D -w/o MMTL -w All Attention | 0.582 | 0.593 | 0.288 | 1.876 | 0.378 | 0.343 |
| 2 | strict | PPBind-3D -w MMTL -w All Attention | 0.617 | 0.618 | 0.374 | 1.779 | 0.383 | 0.343 |
| 3 | strict | PPBind-3D -w MMTL -w S&M Attention | 0.666 | 0.663 | 0.440 | 1.684 | 0.380 | 0.362 |
| 4 | strict | PPBind-1D -w Align -w Unlabeled Samples | 0.648 | 0.631 | 0.403 | 1.690 | 0.004 | 0.004 |
| 5 | strict | PPBind-1D -w Align -w/o Unlabeled Samples | 0.626 | 0.606 | 0.311 | 1.847 | 0.004 | 0.004 |
| 6 | strict | PPBind-1D -w/o Align -w/o Unlabeled Samples | 0.594 | 0.587 | 0.229 | 1.932 | 0.062 | 0.050 |
| 7 | random | PPBind-3D -w MMTL -w S&M Attention | 0.887 | 0.882 | 0.785 | 0.898 | 0.634 | 0.607 |
| 8 | random | PPBind-1D -w Align -w Unlabeled Samples | 0.865 | 0.857 | 0.745 | 0.966 | 0.336 | 0.319 |
| 9 | random | PPBind-1D -w Align -w/o Unlabeled Samples | 0.876 | 0.866 | 0.763 | 0.908 | 0.443 | 0.412 |
| 10 | random | PPBind-1D -w/o Align -w/o Unlabeled Samples | 0.868 | 0.862 | 0.748 | 0.958 | 0.463 | 0.441 |

Table 3: Comparative Information Table for Case 1. 5TRU, 6RP8 has been deleted and does not appear in the case 1 final result.

| Case 1 | | Training Data | | IDist |
|---|---|---|---|---|
| PDB | description | PDB | description | distance |
| 1I85 | Crystal Structure Of The Ctla-4/B7-2 Complex | 1I8L | Human B7-1/Ctla-4 Co-Stimulatory Complex | 0.059 |
| 4ZQK | Structure of the complex of human programmed death-1 (PD-1) and its ligand PD-L1. | 4C9B | Crystal structure of eIF4AIII-CWC22 complex | 0.038 |
| 5B8C | High resolution structure of the human PD-1 in complex with pembrolizumab Fv | 6J6Y | FGFR4 D2 - Fab complex | 0.046 |
| 5GGS | PD-1 in complex with pembrolizumab Fab | 5D8J | Development of a therapeutic monoclonal antibody targeting secreted aP2 to treat type 2 diabetes. | 0.050 |
| 5GGT | PD-L1 in complex with BMS-936559 Fab | 5DWU | Beta common receptor in complex with a Fab | 0.052 |
| 5GGV | CTLA-4 in complex with tremelimumab Fab | 5KVF | Zika specific antibody, ZV-64, bound to ZIKA envelope DIII | 0.063 |
| 5JXE | Human PD-1 ectodomain complexed with Pembrolizumab Fab | 1YQV | The crystal structure of the antibody Fab HyHEL5 complex with lysozyme at 1.7A resolution | 0.046 |
| 5TRU | Structure of the first-in-class checkpoint inhibitor Ipilimumab bound to human CTLA-4 | 5TRU | Structure of the first-in-class checkpoint inhibitor Ipilimumab bound to human CTLA-4 | 0.000 |
| 6RP8 | Crystal Structure of Ipilimumab Fab complexed with CTLA-4 at 2.6A resolution | 5TRU | Structure of the first-in-class checkpoint inhibitor Ipilimumab bound to human CTLA-4 | 0.023 |
| 6XY2 | Crystal structure of CTLA-4 complexed with the Fab of HL32 antibody | 1FE8 | Crystal Structure Of The Von Willebrand Factor A3 Domain In Complex With A Fab Fragment Of Igg Ru5 That Inhibits Collagen Binding | 0.061 |
| 7CGW | Complex structure of PD-1 and tislelizumab Fab | 5K59 | Crystal structure of LukGH from Staphylococcus aureus in complex with a neutralising antibody | 0.050 |
| 8HIT | Crystal structure of anti-CTLA-4 humanized IgG1 MAb–JS007 in complex with human CTLA-4 | 6P67 | Crystal Structure of a Complex of human IL-7Ralpha with an anti-IL-7Ralpha 2B8 Fab | 0.048 |

Table 4: Comparative Information Table for Case 2

| Case 2 | | Training Data | | IDist |
|---|---|---|---|---|
| ID | description | PDB | description | Distance |
| M1 | Anti-Hen Egg lysozyme antibodies | 4PGJ | Human heavy-chain domain antibody in complex with hen egg-white lysozyme | 0.039 |
| M2 | | 4ML7 | Crystal structure of Brucella abortus PliC in complex with human lysozyme | 0.051 |
| M3 | | 1PVH | Crystal structure of leukemia inhibitory factor in complex with gp130 | 0.072 |
| M4 | | 3U7Y | Structure of NIH45-46 Fab in complex with gp120 of 93TH057 HIV | 0.046 |
| M5 | | 1FSK | Complex Formation Between A Fab Fragment Of A Monoclonal Igg Antibody And The Major Allergen From Birch Pollen Bet V 1 | 0.044 |
| M6 | | 4GN4 | OBody AM2EP06 bound to hen egg-white lysozyme | 0.045 |
| M7 | | 5J7C | A picomolar affinity FN3 domain in complex with hen egg-white lysozyme | 0.050 |
| M8 | | 4CJ2 | Crystal structure of HEWL in complex with affitin H4 | 0.054 |
| M9 | | 4PGJ | Human heavy-chain domain antibody in complex with hen egg-white lysozyme | 0.047 |
| M10 | | 5EZO | Crystal Structure of PfCyRPA in complex with an invasion-inhibitory antibody Fab | 0.051 |
| M11 | | 3VG9 | Crystal structure of human adenosine A2A receptor with an allosteric inverse-agonist antibody at 2.7 A resolution | 0.042 |
| M12 | | 4PGJ | Human heavy-chain domain antibody in complex with hen egg-white lysozyme | 0.042 |
| M13 | | 2C1T | Structure of the Kap60p:Nup2 complex | 0.058 |
| M14 | | 4MAY | Crystal structure of an immune complex | 0.047 |
| M15 | | 4CJ0 | Crystal structure of CelD in complex with affitin E12 | 0.051 |
| M16 | | 4ML7 | Crystal structure of Brucella abortus PliC in complex with human lysozyme | 0.055 |
| M17 | | 3G6D | Crystal structure of the complex between CNTO607 Fab and IL-13 | 0.047 |
| M18 | | 4PGJ | Human heavy-chain domain antibody in complex with hen egg-white lysozyme | 0.039 |
| M19 | | 4ML7 | Crystal structure of Brucella abortus PliC in complex with human lysozyme | 0.050 |
| M23 | | 4GN4 | OBody AM2EP06 bound to hen egg-white lysozyme | 0.050 |

Table 5: Case 2 Comparison Information Table Continued

| Case 2 | | Training Data | | IDist |
|---|---|---|---|---|
| ID | description | PDB | description | Distance |
| C1 | | 4ZS7 | Structural mimicry of receptor interaction by antagonistic IL-6 antibodies | 0.058 |
| C2 | | 3KV4 | Structure of PHF8 in complex with histone H3 | 0.068 |
| C3 | | 1VEU | Crystal structure of the p14/MP1 complex at 2.15 A resolution | 0.049 |
| C4 | | 3IDY | Crystal structure of HIV-gp120 core in complex with CD4-binding site antibody b13, space group C2221 | 0.044 |
| C5 | Anti-Hen Egg lysozyme antibodies | 3PL6 | Structure of Autoimmune TCR Hy.1B11 in complex with HLA-DQ1 and MBP 85-99 | 0.043 |
| C6 | | 4ML7 | Crystal structure of Brucella abortus PliC in complex with human lysozyme | 0.047 |
| C7 | | 3T2N | Human hepsin protease in complex with the Fab fragment of an inhibitory antibody | 0.034 |
| C8 | | 3FFC | Crystal Structure of CF34 TCR in complex with HLA-B8/FLR | 0.070 |
| F1 | | 1KIR | Fv Mutant Y(A 50)S (Vl Domain) Of Mouse Monoclonal Antibody D1.3 Complexed With Hen Egg White Lysozyme | 0.050 |
| F2 | | 4GLV | OBody AM3L09 bound to hen egg-white lysozyme | 0.050 |
| F3 | | 1B3S | Structural Response To Mutation At A Protein-Protein Interface | 0.052 |
| F4 | | 3T2N | Human hepsin protease in complex with the Fab fragment of an inhibitory antibody | 0.046 |
| F5 | | 1DZB | Crystal structure of phage library-derived single-chain Fv fragment 1F9 in complex with turkey egg-white lysozyme | 0.043 |
| F6 | | 4GN4 | OBody AM2EP06 bound to hen egg-white lysozyme | 0.051 |
| F7 | | 4ML7 | Crystal structure of Brucella abortus PliC in complex with human lysozyme | 0.045 |
| F8 | | 4PGJ | Human heavy-chain domain antibody in complex with hen egg-white lysozyme | 0.043 |
| F9 | | 1KIR | Fv Mutant Y(A 50)S (Vl Domain) Of Mouse Monoclonal Antibody D1.3 Complexed With Hen Egg White Lysozyme | 0.048 |
| F10 | | 3T2N | Human hepsin protease in complex with the Fab fragment of an inhibitory antibody | 0.036 |
| M19 | | 4ML7 | Crystal structure of Brucella abortus PliC in complex with human lysozyme | 0.050 |
| M23 | | 4GN4 | OBody AM2EP06 bound to hen egg-white lysozyme | 0.050 |

Table 6: Comparative Information Table for Case 3

| Case 3 | | Training Data | | IDist |
|---|---|---|---|---|
| ID | description | PDB | description | Distance |
| VHH1 | | 6CDO | Structure of vaccine-elicited HIV-1 neutralizing antibody vFP16.02 in complex with HIV-1 fusion peptide residue 512-519 | 0.051 |
| VHH2 | | 6UMT | High-affinity human PD-1 PD-L2 complex | 0.037 |
| VHH4 | | 5FUG | Crystal structure of a human YL1-H2A.Z-H2B complex | 0.055 |
| VHH6 | Anti-PD-L1 antibodies | 4I0C | The structure of the camelid antibody cAbHuL5 in complex with human lysozyme | 0.041 |
| VHH9 | | 4JLR | Crystal structure of a designed Respiratory Syncytial Virus Immunogen in complex with Motavizumab | 0.059 |
| VHH10 | | 1DHK | Structure Of Porcine Pancreatic Alpha-Amylase | 0.049 |
| VHH13 | | 1VEU | Crystal structure of the p14/MP1 complex at 2.15 A resolution | 0.053 |
| VHH14 | | 4AYD | Structure of a complex between CCPs 6 and 7 of Human Complement Factor H and Neisseria meningitidis FHbp Variant 1 R106A mutant | 0.040 |
| VHH15 | | 4ML7 | Crystal structure of Brucella abortus PliC in complex with human lysozyme | 0.052 |
| VHH16 | | 4P5T | 14.C6 TCR complexed with MHC class II I-Ab/3K peptide | 0.053 |
| VHH17 | | 3K2M | Crystal Structure of Monobody HA4/Abl1 SH2 Domain Complex | 0.056 |
| VHH18 | | 5GTB | crystal structure of intermembrane space region of the ARC6-PDV2 complex | 0.054 |
| VHH19 | | 1EFN | Hiv-1 Nef Protein In Complex With R96I Mutant Fyn Sh3 Domain | 0.057 |
| VHH20 | | 5E3E | Crystal structure of CdiA-CT/CdiI complex from Y. kristensenii 33638 | 0.041 |
| VHH21 | | 3CHW | Complex of Dictyostelium discoideum Actin with Profilin and the Last Poly-Pro of Human VASP | 0.053 |
| VHH22 | | 6FQ0 | Crystal structure of the CsuC-CsuA/B chaperone-subunit preassembly complex of the archaic chaperone-usher Csu pili of Acinetobacter baumannii | 0.050 |

