# OpenReview forum: "Enhancing PPB Affinity Prediction through Data Integration and Feature Alignment: Approaching Structural Model Performance with Sequences"
_ICLR.cc/2025/Conference — Submitted to ICLR 2025_

### Official Review · Reviewer_msTs · 2024-10-18

**Soundness:** 2
**Presentation:** 2
**Contribution:** 2
**Rating:** 3
**Confidence:** 4

**Summary:**

The paper introduces two new models for the prediction of protein-protein binding affinity (dG) based on protein sequence or protein structure respectively. The data leakage issue for protein-protein interaction datasets is discussed.

**Strengths:**

1. The idea with the alignment of the structure-based and the sequence-based model is interesting.

2. The focus on dealing with the data leakage issue in PPI datasets is appreciated. Figure 4 illustrates the problem with data splitting well.

**Weaknesses:**

1. No benchmarking against other methods. In Related work, there is paragraph listing several methods for dG prediction such as dG-Affinity, PPI-Affinity and AREA-Affinity but none of the tools is benchmarked against in the paper. The authors should (i) make sure the related work is up to date and there are no more recent methods for the PPB affinity prediction task and (ii) authors should compare their methods against SOTA.

2. The "monotonic neural network-constrained multi-task learning (MMRL)" method is not clear at all. What is the operator $M_{\theta_t}$? It just says it as a monotonic neural network. What is the architecture? What are its parameters trained on? Is it trained together with the task for which equation (6) is used as the objective? Can the authors comment on what are the implications of cooptimizing the task and its learning objective?


3. The claimed novelty in partitioning the dataset with iDist is not novel, it has already been done by the authors of iDist [1]. If the authors want to claim novelty, they should explain what is the novelty with respect to [1].

4. Significant part of results for PPBind-1D is missing because the Figure 5 is a duplicate of Figure 4. Please fix.

**Questions:**

1. Did the authors check for potential overlap (e.g. with iDist) between DIPS-plus and PPBAffinity dataset? If there is some overlap, this might explain the success of the alignment procedure.

2. The authors are using DIPS-Plus dataset, which has already been shown to contain many near-duplicates [1]. I suggest using datasets which improved on this issue, are bigger and of higher quality, such as PPIRef [1] or PINDER [2].

3. What is the purpose of Figure 2? It is not well described and it is not sure whether it conveys some important information. Consider removing the figure or explaining its purpose.

4. Line 178-179 "Euclidean distances between each fold of data were calculated as shown." Where is it shown? Please explain in detail (SuppMat can be used).

5. Is the geometric encoder (equations 1-5) completely novel? Did the authors draw inspiration from some existing work? Relevant work should be cited in case inspiration was taken from the literature. If the method is novel, it deserves more attention and it should be discussed in more detail to provide intuition for the equations.

6. I am not sure how advanced is the benchmarking for the dG prediction task, if it is hard to benchmark on that task, the authors could consider using PPI datasets such as PPIRef or PINDER and use the number of contacts between proteins as a proxy for binding affinity [3].

References:

[1] Bushuiev, A., et al. (2024). Learning to design protein-protein interactions with enhanced generalization, ICLR 2024

[2] Kovtun et al. (2024) PINDER: The protein interaction dataset and evaluation resource, bioRxiv 2024.07.17.603980; doi: https://doi.org/10.1101/2024.07.17.603980

[3] Anna Vangone Alexandre MJJ Bonvin (2015) Contacts-based prediction of binding affinity in protein–protein complexes eLife 4:e07454.
https://doi.org/10.7554/eLife.07454

---

> ### Author Response · Authors · 2024-11-25
>
> Thank for your recognition and criticism of our work, as well as for your guidance. The following is our response to the Weaknesses and Questions section.
>
> W1+Q6: **We compared and outperformed multiple algorithms, as detailed in section 5.1 and Table 1 of the paper.**
>
> W2:I'm sorry we didn't explain this part clearly. The operator $M_{\theta}$ is a Monotonic Neural Networks (MNN), which fundamentally represent a monotonic positive function $y = F(x, \theta)$ between two scalar values $x$ and $y$. Antoine,W., et al[2]. constructed such a function by integrating a strictly positive derivative $f(t, \theta)$:
> $F(x, \theta) = \int_0^x f(t, \theta) \, dt + F(0, \theta)$. Here, $F(0, \theta)$ is a constant. We represent $f(t, \theta)$ using a simple MLP, ensuring that the output remains positive by applying a ELU activation function in the final layer and adding one to the network output value. More details refer to APPENDIX A.4.
>
> Indeed, the parameters of the MMN network are trained jointly with those of the PPBind model, as the MMN constraint serves to suppress the monotonic correlation between affinity predictions and ∆G predictions.
>
> **Special reminder, we have renamed Monotonic Neural Networks to $F_{\theta}$, and the original $M_{\theta}$  was defined as the PPBind model**
>
> W3: Yes, Bushuiev, A., et al. introduced the iDist method for PPI (Protein-Protein Interaction) similarity assessment, which exhaustively mines all 202,380 entries from the Protein Data Bank as of June 20, 2023, resulting in a final non-redundant dataset named PPIRef. The authors then pre-trained a model named PPIformer on 3D PPIs from the PPIRef dataset and subsequently fine-tuned it on the SKEMPI V2.0 dataset to predict ΔΔG values. However, we noticed that the authors did not use iDist to partition the dataset; instead, they based their splits on the existing "Hold out proteins" markers in the SKEMPI V2.0 dataset.
> Since the data we used not only included Skempiv2.0, we used iDist to represent the protein complexes and constructed a graph by nearest neighbours (shown in Fig. 2A), and partitioned the data by a graph slicing algorithm called METIS[1], which optimizes the quality of the partitioning by minimizing the edge cuts, and allows us to partition the points that are more similar to each other with Bushuiev, A., et al. is not entirely consistent.
>
> W4: Thank you for pointing out this issue. We have updated the image.
>
> Q1:Yes, there is a certain overlap between DIPS-plus and PPB-Affinity, nevertheless, due to the fact that the DIPS-plus dataset is unlabelled, it can't expose the labels of the test set.
>
> Q2: Many thanks to the reviewers for their valuable suggestions, the PPBRef dataset and the PINDER dataset contain a large amount of high-quality PPI data, which is helpful for our model training, and we will subsequently incorporate these two datasets to train our models.
>
> Q3:Figure 2 is a visualisation of the data partition.Nodes lacking edge connections constitute the "ring" in the figure. Conversely, nodes with a greater number of edge connections will be situated in closer proximity to the "centre" of the circle. We use different colours to represent different data folds, and it is evident that each data fold exhibits distinct characteristics. Optimising the partitioning quality through minimising edge cutting by METIS[1], it is possible to group together nodes with greater similarity. This approach to data partitioning facilitates the enrichment of homologous or similar structures within a single data fold. In order to make the meaning of Figure 2 clear to all, we have added some notes in the article.
>
> Q4:Sorry for not writing this part clearly. It is shown in Figure 9(in Appendix A.3) in the latest paper, and our paper has also been revised.
>
> Q5:Thanks for the reminder, yes, this part of the work was inspired by DDG-Pred[3] and RDE-network[4], based on which the SELF' and “MUTUAL” modes were proposed to improve the performance of the model.We also pointed out this point in the article.
>
> References:
>
> [1]Karypis, George and Vipin Kumar. “A Fast and High Quality Multilevel Scheme for Partitioning Irregular Graphs.” SIAM J. Sci. Comput. 20 (1998): 359-392.
>
> [2]Wehenkel, Antoine and Louppe, Gilles, Unconstrained monotonic neural networks,  https://doi.org/10.48550/arXiv.1908.05164, NeurIPS 2019
>
> [3]S. Shan, S. Luo, Z. Yang, J. Hong, Y. Su, F. Ding, L. Fu, C. Li, P. Chen, J. Ma, X. Shi, Q. Zhang, B. Berger, L. Zhang, J. Peng, Deep learning guided optimization of human antibody against SARS-CoV-2 variants with broad neutralization, Proc. Natl. Acad. Sci. U.S.A.119 (11) e2122954119, https://doi.org/10.1073/pnas.2122954119 (2022).
>
> [4]Shitong Luo and Yufeng Su and Zuofan Wu and Chenpeng Su and Jian Peng and Jianzhu Ma, Rotamer Density Estimator is an Unsupervised Learner of the Effect of Mutations on Protein-Protein Interaction, https://doi.org/10.1101/2023.02.28.530137, ICLR203

---

> > ### Comment · Reviewer_msTs · 2024-11-25
> >
> > Thanks to the authors for the response. The Table 1 was not originally present, it was added during rebuttal, right? Please clarify what changes to the PDF were done.
> >
> > Is it really true that authors cannot evaluate the other methods on Test set 1 in Table 1? This manuscript (https://www.biorxiv.org/content/10.1101/2024.03.14.584935v1.full.pdf) has an evaluation of Prodigy, Dfire and other models for "Test set 1" including complexes with 2 or more chains.
> >
> > I went over the PDF again and it is really hard to understand the paper and its contributions and how they improve over others. If the authors are confident about their method, they should work on the presentation significantly as current version puts a lot of demand on the reader.
> >
> > I maintain my score of 3.

---

### Official Review · Reviewer_RR8f · 2024-11-01

**Soundness:** 2
**Presentation:** 3
**Contribution:** 2
**Rating:** 5
**Confidence:** 3

**Summary:**

This work addresses the task of predicting protein-protein binding (PPB) affinity to improve the efficiency of high-throughput screening in protein drug development. The motivation behind this work is to overcome limitations associated with traditional laboratory screening methods for PPB affinity, which are costly, time-consuming, and not well-suited for high-throughput applications. Additionally, existing deep learning models often lack sufficient high-quality data or generalization capability due to limited compatibility with diverse affinity data. To accomplish this, the authors developed two AI models, PPBind-3D and PPBind-1D. In this process, they focused on (1) utilizing a novel and large dataset, (2) strictly partitioning data for performance testing, and (3) introducing a "feature alignment" mechanism. The authors demonstrated the performance of their models using the PPB-Affinity dataset and three virtual screening cases.

**Strengths:**

The structure of the article is clear and easy to follow. The figures are well-designed. The authors use biological measurement terminology correctly, such as subscripting characters where appropriate.

**Weaknesses:**

1. Lack of Baselines: Section 2, "Related Work," mentions several existing studies on this task. However, **there is no comparison between the proposed models and previous models**, aside from the comparisons within the authors' own models (1D, 3D, and aligned).
2. Missing Important Figure: **Figure 5 is an exact copy of Figure 4**. While this is likely unintentional, the absence of additional descriptions or tables showing the performance of the 1D model is a significant issue, as the case studies alone cannot demonstrate the model's general performance.
3. Presentation Weaknesses: Tables would be more suitable for displaying model performance, especially for conference papers. Additionally, the title is overly long and lacks focus. For readers unfamiliar with the subject, the abbreviation "PPB" may be confusing; using it in an already lengthy title is somewhat counterintuitive. In Figures 7 and 8, the x-axis ranges in panels A, B, and C are inconsistent, making it difficult to identify trends. There is also a typo in line 413: "affiity."

**Questions:**

1. What is the SOTA performance on the PPB task and on the datasets used, such as PPB-Affinity?
2. Why specifically choose iDist and K-nearest neighbors methods? Although PDB code-based and time-based splitting methods are insufficient, why not consider sequence similarity-based splitting methods, for example?
3. As the authors mentioned in Section 3, the lack of strict data splitting is problematic for validating models. Why do the authors use models trained on randomly split data in Section 5.2? How can model performance be validated when the data splitting process is not strict? Will there be a significant drop in model performance in the virtual screening scenario introduced in Section 5.2 when using strictly split datasets?
4. Why was contrastive learning not selected for feature alignment?
5. I'm unsure whether the log₂ enrichment ratio qualifies as an affinity measurement.

---

> ### Author Response · Authors · 2024-11-25
>
> Thanks for your recognition and criticism of our work, as well as for your guidance. The following is our response to the Weaknesses and Questions.
>
> W1-W3: Thanks for pointing out the problems with our presentation. We've updated the paper with the correct Figure 5 and corrected the typos.
>
> Q1: We compared and outperformed multiple algorithms, as detailed in section 5.1 and Table 1 of the paper.
> |  |  | **Test set 1** |  | | **Test set 2** ||
> |:---:|---:|:---:|---|---:|:---:|---|
> |  | Pearson | MAE(kcal/mol) | RMSE(kcal/mol) | Pearson | MAE(kcal/mol) | RMSE(kcal/mol) |
> | PRODIGY | - | - | - | 0.28 | 2.47 | 3.52 |
> | DFIRE | - | - | - | 0.08 | 25.05 | 29.17 |
> | CP\_PIE | - | - | - | -0.10 | 10.90 | 11.27 |
> | ISLAND | - | - | - | 0.28 | 2.30 | 2.85 |
> | PPI-Affinity | - | - | - | 0.49 | 1.83 | 2.40 |
> | ProBAN | 0.60 | 1.60 | 1.95 | 0.55 | 1.75 | 2.28 |
> | PPBind-3D(ours) | 0.626 | 1.482 | 1.898 | 0.559 | 1.647 | 2.210 |
>
> Q2: We consider that in some cases, a considerable portion of the sequence may be redundant. For example, if a ligand protein interacts with the extracellular segment of a receptor protein, the intracellular segment of the receptor protein will not interact with the ligand protein. For AI models, especially those that use structural information, the information of the intracellular segment is redundant. However, the similarity between the complete receptor protein sequence and the sequence of only the extracellular segment of the receptor protein is not high. When dividing data based on sequence similarity, these two homologous datas are likely to be divided into different datasets, resulting in a certain degree of data leakage. And we didn't find a good method to calculate the identity between sequences of complexes.
> Meanwhile, Bushuiev et al. found that partitioning sequences based on sequence similarity leads to a significant proportion of data leakage[1], and the iDist[1] method proposed involves calculating the SE (3) equivariant features of the composite binding interface and evaluating the Euclidean distance to determine the degree of similarity/homology between two composites.
>
> Q3:By strictly dividing the data, we can see the generalisation ability of the model, which shows that our model architecture design and training task are feasible. Under the strict division, there are many types of interfaces that the model has not seen before. Although the model generalises to this part of the data that has not been seen before, it is certainly not as strong as letting the model learn this part of the data directly. Meanwhile, in practical applications, we hope that the model can predict accurately under various kinds of data, therefore, in the three cases, we use the data trained under non-strict division. Although the amount of data is small, the three virtual screening applications  are a direct way to validate the performance of the model under non-strictly divided data. Additionally, we have compared the cases data with the training data, and the details can be found in Appendix A.6.
>
> Q4:We have used contrast learning in our earlier work in conjunction with our proposed method, and found that it is more straightforward and effective to use direction and magnitude of vectors than contrast learning. We suspect that the traditional contrastive learning method of constructing negative samples was not as appropriate for this work, and because the direction and magnitude of vector gave better results, we did not continue to investigate how to use contrast learning better.
>
> Q5:The log2 enrichment ratio used in this work comes from the paper[2]. By deep sequencing ACE2 and screening variants with high binding ability to SARS-CoV-2 spiking proteins, and then comparing the enrichment ratios of different variants, it was found that ACE2 variants with higher binding affinity to S proteins usually have higher enrichment ratios.
>
> References:
>
> [1]Bushuiev, A., et al. (2024). Learning to design protein-protein interactions with enhanced generalization, ICLR 2024, https://arxiv.org/abs/2310.18515
>
> [2]Kui K. Chan et al. ,Engineering human ACE2 to optimize binding to the spike protein of SARS coronavirus 2.Science369,1261-1265(2020).DOI:10.1126/science.abc0870

---

> > ### Comment · Reviewer_RR8f · 2024-11-27
> > **Response to the authors' response**
> >
> > I appreciate the substantial effort the authors have devoted to revising their manuscript. After reviewing their response, I acknowledge the challenges associated with implementing contrastive learning in this specific context and understand their rationale for adopting alternative approaches. I have no further questions regarding these aspects.
> >
> > That said, I must note that the newly presented table and accompanying explanations still lack the clarity and persuasiveness needed to fully support the contributions of this work. While the authors have addressed some of the original weaknesses, the paper remains challenging to understand in certain sections. My initial score of 3 primarily stemmed from the omission of critical tables and figures. Given the authors' revisions and after considering the reviews of other reviewers, I still find this paper below the threshold for recommending acceptance and I am inclined to revise my score to a 5.
> >
> > I look forward to further discussions if additional experimental results or more comprehensive explanations are provided to strengthen this work.

---

### Official Review · Reviewer_cVsB · 2024-11-02

**Soundness:** 3
**Presentation:** 2
**Contribution:** 3
**Rating:** 6
**Confidence:** 3

**Summary:**

This paper presents a new approach for predicting protein-protein binding (PPB) affinity, which is essential in drug discovery. The authors developed two models, PPBind-3D and PPBind-1D.

PPBind-3D leverages structural data to predict affinity using advanced data integration and a multi-task learning approach, which enables it to generalize well despite data variability. PPBind-1D, on the other hand, relies on sequence data alone, making it more applicable when structural data is unavailable.

To align PPBind-1D's performance with that of PPBind-3D, the authors introduced an alignment technique using additional unlabeled data, helping the sequence-based model approximate structural model performance. Evaluations show that these models, particularly PPBind-1D, can support high-throughput screening by predicting PPB affinity accurately, even under strict data partitioning to avoid leakage. The work’s impact lies in enhancing drug discovery workflows with a method that bridges data gaps while maintaining predictive accuracy.

**Strengths:**

The methodology is strong, incorporating strict data partitioning and monotonic multi-task learning to enhance model generalization. While the technical explanations are clear, some sections could benefit from simplification for accessibility. This work’s flexible, scalable model has significant implications for drug discovery, offering a valuable tool for high-throughput screening relevant to both computational biology and AI communities.

**Weaknesses:**

The alignment method for integrating sequence-based features with structure-based predictions is intriguing but not fully detailed. Providing more in-depth explanations and visualizations, especially of how alignment influences the latent spaces between PPBind-1D and PPBind-3D, would strengthen understanding and reproducibility.


While the paper includes ablation studies, adding more direct comparisons with existing models (e.g., CSM-AB, AREA-AFFINITY) using standard benchmarks would clarify the novelty and effectiveness of the proposed approach. Highlighting quantitative gains over established models would emphasize the advantages of PPBind-1D and PPBind-3D.


A brief analysis of feature importance or interpretability of predictions, particularly around how sequence and structural features affect affinity, would make the work more useful for practical applications and provide valuable insights into model behavior.

**Questions:**

Can the authors elaborate on the performance of PPBind-1D and PPBind-3D across additional external datasets? It would be helpful to understand how well these models generalize to datasets beyond PPB-Affinity and DMS-Het, especially for applications with different data types or measurement techniques.


Adding an analysis on which structural or sequence features most impact the predictions would provide valuable insights. Are there any interpretability tools (e.g., SHAP values or feature importance rankings) applied to understand how features contribute to binding affinity predictions? This could increase the model’s practical applicability and user trust.

Can the authors provide more details on the computational resources required for PPBind-3D versus PPBind-1D? A comparison in training and inference time, along with any scalability insights, would help evaluate the model’s applicability in real-world, high-throughput scenarios.

---

> ### Author Response · Authors · 2024-11-25
>
> Thank you for your recognition and criticism of our work, as well as for your guidance. The following is our response to the Weaknesses and Questions section.
>
> W1: To observe the effectiveness of 'Alignment'(see Section 5.2, Figure 5), the feature representations are visualised by dimensionality reduction using the following steps:
> **Step1** Extract high-dimensional feature representations of the training samples using the PPBind-3D model, and fit a dimensionality reduction function $F_{U}$ using the UMAP (Uniform Manifold Approximation and Projection) algorithm. **Step2** Extract high-dimensional feature representations of the validation samples using the PDBind-3D, PDBind-1D, and PPBind-1D-w/o Align models, respectively. **Step3** Individually project the three sets of high-dimensional feature representations onto a 2D plane using the fitted $F_{U}$  function and visualize them.
>
> By comparing the three dimensionality reduction visualizations Fig5, it can be observed that the dimensionality reduction representation of the PPBind-1D model retains a similar data topological structure to that of the PPBind-3D model, whereas the PPBind-1D model without 'Alignment' exhibits a scattered state.
> This indicates that the representations extracted by the PPBind-1D model are similar to those of the PPBind-3D model, suggesting that it is possible to enhance the prediction accuracy of the PPBind-1D model to the level of the PPBind-3D model through our proposed 'Alignment' method.
> We have also added the above content in the article(see Section 5.2).
>
> W2:We compared and outperformed multiple algorithms, as detailed in section 5.1 and Table 1 of the paper.
> |  |  | **Test set 1** |  | | **Test set 2** ||
> |:---:|---:|:---:|---|---:|:---:|---|
> |  | Pearson | MAE(kcal/mol) | RMSE(kcal/mol) | Pearson | MAE(kcal/mol) | RMSE(kcal/mol) |
> | PRODIGY | - | - | - | 0.28 | 2.47 | 3.52 |
> | DFIRE | - | - | - | 0.08 | 25.05 | 29.17 |
> | CP\_PIE | - | - | - | -0.10 | 10.90 | 11.27 |
> | ISLAND | - | - | - | 0.28 | 2.30 | 2.85 |
> | PPI-Affinity | - | - | - | 0.49 | 1.83 | 2.40 |
> | ProBAN | 0.60 | 1.60 | 1.95 | 0.55 | 1.75 | 2.28 |
> | PPBind-3D(ours) | 0.626 | 1.482 | 1.898 | 0.559 | 1.647 | 2.210 |
>
> W3+Q2:  Thank you for your suggestions. Conducting explainability analysis is indeed a meaningful and engaging task, but we have not been able to complete it within our limited timeframe. Perhaps this will be addressed in our upcoming work.
>
> Q3:With batch size=16, using a single NVIDIA A100 GPU, training  PPBind-3D for 100000 iterations takes about 7hours, and training PPBind-1D for 360,000 iterations takes about 22hours. The inference speed of PPB-1D and PPB-3D is approximately 12s and 6s per thousand samples, respectively.

---

### Official Review · Reviewer_7rLy · 2024-11-04

**Soundness:** 2
**Presentation:** 2
**Contribution:** 2
**Rating:** 3
**Confidence:** 3

**Summary:**

This work introduces two AI models called PPBind-3D and PPBind-1D to enhance protein-protein binding (PPB) affinity prediction, which is crucial for protein drug development. In detail, PPBind-3D leverages structural data near binding interfaces, supported by a novel monotonic neural network-based multi-task learning (MMTL) approach, which integrates diverse experimental datasets to improve generalization. Besides, PPBind-1D uses sequence-based data, aligning with structural predictions to address scenarios when structural data is limited. In the experiments, the authors demonstrate the models' potential to support high-throughput virtual screening of PPB affinities by illustrating three case studies in virtual screening applications.

**Strengths:**

This paper introduces a novel approach to protein-protein binding (PPB) affinity prediction, integrating both structural and sequence-based models to address the high-throughput demands of drug discovery. The models, PPBind-3D and PPBind-1D, are designed with a sequence-structure alignment strategy that allows the sequence-only model to gain structural insights indirectly. This innovation effectively bridges the gap where structural data is unavailable. Besides, the authors use a monotonic neural network-based multi-task learning (MMTL) framework to incorporate heterogeneous affinity data, enhancing the model’s robustness while handling variations in measurement types. The authors also pay attention to data partitioning to avoid data leakage. These methodological choices are evaluated by ablation studies and real-world virtual screening case studies.

The clarity of the presentation is overall good to let readers understand the proposed models and the experiments. In terms of impact, this paper addresses a critical challenge in high-throughput screening by providing a flexible solution that has both structural and sequence-based models.

**Weaknesses:**

The baseline models are missing in experiments, so it is unknown how well the proposed models perform when compared to existing ones.

When partitioning the data, the authors only provide the partition performance according to distances. But it is hard to understand what distance level is good or not. I feel that using the protein sequence identity ratio between different proteins can be more straightforward.

For the results of the three cases in virtual screening applications, there can be data leakage between the test set and the training set. It would be interesting to know if the well-predicted structures/sequence data exist in the training set or share high similarities with the data in the training set.

**Questions:**

Since the proposed methods are meant to be applied to virtual screening, what is the efficiency of them? For example, the inference speed and memory consumption.

---

> ### Author Response · Authors · 2024-11-25
>
> Thank you for your recognition and criticism of our work, as well as for your guidance. The following is our response to the Weaknesses and Questions section.
>
> W1: We compared and outperformed multiple algorithms, as detailed in section 5.1 and Table 1 of the paper.
>
> |  |  | **Test set 1** |  | | **Test set 2** ||
> |:---:|---:|:---:|---|---:|:---:|---|
> |  | Pearson | MAE(kcal/mol) | RMSE(kcal/mol) | Pearson | MAE(kcal/mol) | RMSE(kcal/mol) |
> | PRODIGY | - | - | - | 0.28 | 2.47 | 3.52 |
> | DFIRE | - | - | - | 0.08 | 25.05 | 29.17 |
> | CP\_PIE | - | - | - | -0.10 | 10.90 | 11.27 |
> | ISLAND | - | - | - | 0.28 | 2.30 | 2.85 |
> | PPI-Affinity | - | - | - | 0.49 | 1.83 | 2.40 |
> | ProBAN | 0.60 | 1.60 | 1.95 | 0.55 | 1.75 | 2.28 |
> | PPBind-3D(ours) | 0.626 | 1.482 | 1.898 | 0.559 | 1.647 | 2.210 |
>
> W2: We consider that in some cases, a considerable portion of the sequence may be redundant. For example, if a ligand protein interacts with the extracellular segment of a receptor protein, the intracellular segment of the receptor protein will not interact with the ligand protein. For AI models, especially those that use structural information, the information of the intracellular segment is redundant. However, the similarity between the complete receptor protein sequence and the sequence of only the extracellular segment of the receptor protein is not high. When dividing data based on sequence similarity, these two homologous samples are likely to be divided into different datasets, resulting in a certain degree of data leakage. And we didn't find a good method to calculate the identity between complexes, so we didn't use sequence similarity to partition the data.
> Meanwhile, Bushuiev et al. found that partitioning sequences based on sequence similarity leads to a significant proportion of data leakage[1], and the iDist[1] method proposed involves calculating the SE (3) equivariant features of the composite binding interface and evaluating the Euclidean distance to determine the degree of similarity/homology between two composites.
>
> W3:Thanks for your reminder. At the sequence level, we are unable to provide specific numerical values to demonstrate the absence of overlap, as we are unsure of the degree of sequence similarity between complexes. However, at the structural level, we used iDist[1] to represent all complexes in the three cases and compared them with the training set.
> We find that one sample in case1 appears in the training set (PDB ID: 5TRU), and that one sample in case1 is highly similar to 5TRU in the training set (PDB ID: 6RP8). We removed these two samples and recalculated the correlation metrics and found that the correlation metrics did not decrease significantly due to the removal of these two samples, which we believe further validates the excellence of our model. We have included the structural comparison analysis of the three cases with the training data in the appendix, hoping to address your concern.
>
> Q1: The inference speed is not the bottleneck of high-throughput screening. On one A100 GPU, the inference speed of PPB-1D and PPB-3D is approximately 12s and 6s per thousand samples, respectively. Our proposed method is mainly intended to solve the problem of predicting the strength of affinity more accurately in practical applications where structural information is not available and only sequence information is available.
>
> References :
>
> [1]Bushuiev, A., et al. (2024). Learning to design protein-protein interactions with enhanced generalization, ICLR 2024, https://arxiv.org/abs/2310.18515

---

### Official Review · Reviewer_kvVk · 2024-11-04

**Soundness:** 3
**Presentation:** 3
**Contribution:** 3
**Rating:** 6
**Confidence:** 4

**Summary:**

In this paper, PPBind-3D and PPBind-1D are developed to predict protein-protein binding affinity based on three datasets PPB-Affinity dataset, Heterogeneous Affinity Dataset and DIPS-Plus dataset. PPBind-3D used SE(3)-Invariant attention module to capture structural information near the protein-protein binding interface to make its predictions.  PPBind-1D was developed using sequence data to address the lack of structural data in practical applications.

**Strengths:**

The use of monotonic neural network-constrained multi-task learning (MMTL) expanded the development dataset to over 23,000 samples and helped to improve the model’s generalization abilities.

SE(3)-Invariant attention is used to get features of protein complex structures using the iDist algorithm, and then clustering the protein complex structure features based on graph partition algorithms helps to address the data leakage problem.

**Weaknesses:**

Currently, the code doesn’t contain a data partition process.

The paper would be better if including other methods to compare their performance with PPBind-3D.

The metrics used to estimate performance only include spearman or Pearson correlation, lack of RMSE.

**Questions:**

In Section 3.1, what is the reference paper for PPB-Affinity Dataset?

In Section 4.1, you set the distance threshold for identifying protein-binding interface amino acids as 8 Å between the C-alpha atoms of two amino acids. This choice may seem somewhat arbitrary; could you elaborate further on the rationale behind selecting 8 Å as the threshold?

Could you explain more about the data partition process, and why it can help to solve the data leakage problem?

---

> ### Author Response · Authors · 2024-11-25
>
> Thank you for your recognition and criticism of our work, as well as for your guidance. The following is our response to the Weaknesses and Questions section.
>
> W1：The data splitting part is already included in the code we provide, please refer to the https://anonymous.4open.science/r/PPBind-for-ICLR2025/PPI_split/calculate_interface_embedding.py
>
> W2/3：We compared and outperformed multiple algorithms, as detailed in section 5.1 and Table 1 of the paper.Evaluation metrics include MAE and RMSE.
>
> |  |  | **Test set 1** |  | | **Test set 2** ||
> |:---:|---:|:---:|---|---:|:---:|---|
> |  | Pearson | MAE(kcal/mol) | RMSE(kcal/mol) | Pearson | MAE(kcal/mol) | RMSE(kcal/mol) |
> | PRODIGY | - | - | - | 0.28 | 2.47 | 3.52 |
> | DFIRE | - | - | - | 0.08 | 25.05 | 29.17 |
> | CP\_PIE | - | - | - | -0.10 | 10.90 | 11.27 |
> | ISLAND | - | - | - | 0.28 | 2.30 | 2.85 |
> | PPI-Affinity | - | - | - | 0.49 | 1.83 | 2.40 |
> | ProBAN | 0.60 | 1.60 | 1.95 | 0.55 | 1.75 | 2.28 |
> | PPBind-3D(ours) | 0.626 | 1.482 | 1.898 | 0.559 | 1.647 | 2.210 |
>
> Q1:PPB-Affinity is not yet in press, and will be published in 'Nature Scientific Data' in December.
>
> Q2:We are sorry to say that after rechecking our code, we found that 8Å was the threshold used in the old version of our code, and in the latest version, we used a threshold of 10Å. The reason why we used 10Å was because AlphaFold3[1] used 15Å as the threshold for calculating iLDDT, and we considered that the strength of the affinity is strongly correlated with the residue contact distance, so we set the 10Å threshold to ensure that the interface is more accurate.
>
> Q3:Our proposed PPBind-3D is based on the complex's interface for affinity prediction.In order to distinguish the difference of interface, Bushuiev, A. et al[2] designed an SE(3)-invariant interface characterisation algorithm,**iDist**,which can represent the  interface as a vector, and the **PROTEIN-PROTEIN INTERFACE DEDUPLICATION** can be well accomplished by using the Euclidean distance. The combination of iDist and the graph partitioning algorithm **METIS** [3] for data partition not only effectively avoids the appearance of same-source interfaces in different groups, but also clusters more similar interfaces in the same group, which effectively avoids data leakage that leads to the incorrect evaluation of the performance of PPbind-3D.
>
> References:
>
> [1]Abramson, J., Adler, J., Dunger, J. et al. Accurate structure prediction of biomolecular interactions with AlphaFold 3. Nature 630, 493–500 (2024). https://doi.org/10.1038/s41586-024-07487-w
>
> [2]Bushuiev, A., et al. (2024). Learning to design protein-protein interactions with enhanced generalization, ICLR 2024, https://arxiv.org/abs/2310.18515
>
> [3]Karypis, George and Vipin Kumar. “A Fast and High Quality Multilevel Scheme for Partitioning Irregular Graphs.” SIAM J. Sci. Comput. 20 (1998): 359-392.

---

### Author Response · Authors · 2024-11-26

We have made modifications and additions to the paper, highlighting the differences from the previous version in blue. The main changes are as follows:

1. **Section 3.2 DATA PARTITIONING**:
   - Provided a brief supplement to the method and an explanation of Figure 2.
   - Moved the comparison of different methods to APPENDIX A.4 (COMPARE DIFFERENT WAYS OF PARTITIONING DATA).
2. **Section 4.1 PPBIND-3D**:
   - Added the sources of inspiration for our model (DDG-pred[1] & RDE-Network[2]).
   - Optimized Equation 6 and further explained the meanings of the various symbols.
3. **Section 5.1 EVALUATION**:
   - Corrected a error where the scatter plots for PPBind-3D and PPBind-1D were identical; we have updated the plot for PPBind-1D.
   - Included comparisons with various baseline models, presenting the metrics in Table 1.
4. **Section 5.1 VISUALIZATION OF ALIGNMENT**:
   - This section is newly added to further explain the impact of "Alignment". We performed dimensionality reduction visualization on high-dimensional feature vectors to support the benefits of "Alignment".
5. **Section 5.3**:
   - Following the reviewers' feedback, we identified two unreasonable samples in case 1. After removing these samples, we redrew the corresponding scatter plot.
   - Additionally, we conducted a comparative analysis of the samples from the three cases against the training data, detailed in APPENDIX A.6 (COMPARISON OF CASE DATA).
6. **APPENDIX A.2 DETAILS ON THE HYPERPARAMETERS**:
   - Supplemented information regarding the training time required for the model.
7. **APPENDIX A.3 COMPARE DIFFERENT WAYS OF PARTITIONING DATA**:
   - Compared different methods of data partitioning and provided a brief analysis.
8. **APPENDIX A.4 DETAILS OF MONOTONIC CONTROL**:
   - The original section on monotonic neural networks was somewhat vague, so we have elaborated further in A.4.
9. **APPENDIX A.6 (COMPARISON OF CASE DATA)**:
   - Conducted a comparative analysis of the samples from the three cases against the training data, with the results presented in Tables 3-6.

We hope these revisions enhance the clarity and quality of our paper.

References:

[1]S. Shan, S. Luo, Z. Yang, J. Hong, Y. Su, F. Ding, L. Fu, C. Li, P. Chen, J. Ma, X. Shi, Q. Zhang, B. Berger, L. Zhang, J. Peng, Deep learning guided optimization of human antibody against SARS-CoV-2 variants with broad neutralization, Proc. Natl. Acad. Sci. U.S.A.119 (11) e2122954119, https://doi.org/10.1073/pnas.2122954119 (2022).

[2]Shitong Luo and Yufeng Su and Zuofan Wu and Chenpeng Su and Jian Peng and Jianzhu Ma, Rotamer Density Estimator is an Unsupervised Learner of the Effect of Mutations on Protein-Protein Interaction, https://doi.org/10.1101/2023.02.28.530137, ICLR203

---

### Meta-Review · Area_Chair_4c3i · 2024-12-17

**Metareview:**

This paper proposes two new AI models (PPBind-3D and PPBind-1D) for predicting protein-protein binding (PPB) affinity, a key step in drug discovery. One model uses 3D structural data and applies monotonic neural networks and multi-task learning to handle diverse experimental datasets; the other relies on 1D sequence data and uses an alignment mechanism to approximate the performance of the 3D model when structural data are unavailable. The authors also emphasize a careful data partitioning strategy to avoid data leakage, and they present case studies in virtual screening scenarios.

First of all, let me thank the authors -- the authors actively engaged with the reviewers and provided additional experiments, tables, and some clarifications during the rebuttal. They tried to fix missing figures, added baseline comparisons, and included evaluation metrics like MAE and RMSE. However, reviewers still had issues even after revisions. While some reviewers softened their view slightly (from 3 to 5), none expressed strong enthusiasm for acceptance after the rebuttal period. The reviewers remained concerned about clarity, limited benchmarking with external methods, and lingering complexity in understanding the method and the data partition steps.

Common concerns raised by reviewers: (1) The need for a more comprehensive set of baselines from the literature, since referencing existing methods alone isn't enough.
(2) The confusion around the monotonic network-based multi-task learning part and how exactly it’s integrated.
Also, multiple reviewers questioned the clarity of the data partitioning and the alignment mechanisms. While authors tried to improve clarity, reviewers still felt the paper's presentation and explanations were not fully adequate.

Although with huge potential. it is current form may be not ready for acceptance.

**Additional Comments On Reviewer Discussion:**

see above

---

### Decision · Program_Chairs · 2025-01-22

Reject